# Different dendritic domains of the GnRH neuron underlie the pulse and surge modes of GnRH secretion in female mice

Li Wang[1†], Wenya Guo[1†], Xi Shen[1†], Shel Yeo[2,3†], Hui Long[1†], Zhexuan Wang[4], Qifeng Lyu[1]*, Allan E Herbison[2,3]*, Yanping Kuang[1]*

[1]Department of Assisted Reproduction, Shanghai Ninth People's Hospital, Shanghai JiaoTong University School of Medicine, Shanghai, China; [2]Centre for Neuroendocrinology, Department of Physiology, University of Otago, Dunedin, New Zealand; [3]Department of Physiology, Development and Neuroscience, University of Cambridge, Cambridge, United Kingdom; [4]School of Basic Medical Sciences, Institutes of Brain Science, Fudan University, Shanghai, China

**\*For correspondence:**
lyuqifeng@126.com (QL);
aeh36@cam.ac.uk (AEH);
kuangyanp@126.com (YK)

[†]These authors contributed equally to this work

**Competing interests:** The authors declare that no competing interests exist.

**Abstract** The gonadotropin-releasing hormone (GnRH) neurons exhibit pulse and surge modes of activity to control fertility. They also exhibit an unusual bipolar morphology comprised of a classical soma-proximal dendritic zone and an elongated secretory process that can operate as both a dendrite and an axon, termed a 'dendron'. We show using expansion microscopy that the highest density of synaptic inputs to a GnRH neuron exists at its distal dendron. In vivo, selective chemogenetic inhibition of the GnRH neuron distal dendron abolishes the luteinizing hormone (LH) surge and markedly dampens LH pulses. In contrast, inhibitory chemogenetic and optogenetic strategies targeting the GnRH neuron soma-proximal dendritic zone abolish the LH surge but have no effect upon LH pulsatility. These observations indicate that electrical activity at the soma-proximal dendrites of the GnRH neuron is only essential for the LH surge while the distal dendron represents an autonomous zone where synaptic integration drives pulsatile GnRH secretion.

## Introduction

The gonadotropin-releasing hormone (GnRH) neurons represent the final output cells of a complex neuronal network that integrates a wide variety of internal and external factors to control the fertility of the individual (*Herbison, 2016*). In females, these neurons secrete GnRH into the pituitary portal circulation either as discrete pulses approximately once every hour or as a prolonged surge at the mid-point of the ovarian cycle to initiate ovulation (*Goodman, 2015*; *Karsch et al., 1997*; *Levine, 2015*).

The GnRH neurons exhibit a bipolar morphology with synaptic inputs clustered around the soma and proximal dendrites (*Campbell et al., 2005*; *Li et al., 2016*; *Moore et al., 2015*) and the action potential initiation site located 50–100 µm down one of the dendritic processes (*Herde and Herbison, 2015*; *Iremonger and Herbison, 2012*; *Roberts et al., 2008*). To date, synaptic integration within this soma-dendritic zone of the GnRH neuron has been the prime focus of attention in understanding how these cells generate pulse and surge modes of GnRH release. However, recent studies in the mouse have highlighted that one or both of the dendritic processes arising from a GnRH neuron can project for over 4000 µm before terminating in short neurosecretory axons within the secretory zone of the median eminence (*Herde et al., 2013*; *Moore et al., 2018b*). Unusually, these processes conduct action potentials from the soma-dendritic zone but also receive synaptic inputs, with their blended dendritic/axonal properties leading them to be termed 'dendrons' (*Herde et al., 2013*). These morphological studies have raised the possibility that synaptic inputs directed at GnRH

neuron distal dendrons immediately adjacent to the median eminence secretory zone play a role in regulating GnRH secretion. The present studies were aimed at examining the significance and physiological relevance of the GnRH neuron distal dendron in vivo.

## Results

### Synaptic densities at the soma-dendritic and distal dendron zones of the GnRH neuron

While it has been possible to observe synapses on the GnRH neuron distal dendron using electron microscopy (*Moore et al., 2018b*), their potential significance is unknown. To examine the relative density of synaptic inputs at the dendron compared with more proximal compartments of the GnRH neuron, we used expansion microscopy (*Chen et al., 2015*; *Chozinski et al., 2016*) to generate high-resolution images of synaptophysin-immunoreactive terminals apposing GnRH neurons in adult diestrous female GnRH-GFP mice (N = 4)(*Figure 1*). Expansion microscopy operates on the principal of isotropic swelling of the tissue specimen (typically by 4–5 times) before undertaking regular confocal microscopy; thereby providing ~70 nm resolution in brain tissue (*Wassie et al., 2019*). Although synaptophysin can be used to reliably mark almost all presynaptic terminals, a global post-synaptic marker is not available. Hence, to be able to define a synapse using only presynaptic and cytoplasmic labels, we first visualized $GABA_A$ receptors on GnRH-GFP neurons by labelling for GFP alongside the pre- and post-synaptic markers vesicular GABA transporter (VGAT) and gephyrin, respectively (*Figure 1—figure supplement 1*). We analyzed 25 'side-on' profiles where VGAT and gephyrin were opposed to one another, thereby, defining a synapse (*Figure 1—figure supplement 1A*). This revealed that the overlap between cytoplasmic GFP and the presynaptic marker VGAT must be >0.23 μm (0.95 μm post-expansion) to represent a synapse; the gephyrin signal was contained within the GFP signal and all 25 synapses had an overlap in VGAT and GFP signals > 0.23 μm (*Figure 1—figure supplement 1A*). This value (0.23 μm) does not reflect the size of the synaptic cleft but rather the minimum overlap between the irregularity of pre- and cytoplasmic compartments at the synapse when viewed in 2D with this resolution. As not all appositions can be viewed in a 'side-on' profile, we also examined appositions in the 'face view' orientation that require scanning in the z-axis (*Figure 1—figure supplement 1B*). Intensity profiles from a total of 12 GABAergic synapses were plotted with synapses found to have >0.42 μm (1.75 μm post-expansion) overlap in VGAT and GFP signals in the z-axis (*Figure 1—figure supplement 1B*). These observations set the parameters for determining synapses on GnRH-GFP neurons using the global presynaptic marker synaptophysin and GFP as the cytoplasmic label.

In agreement with electron microscopic studies (*Kasthuri et al., 2015*), only a very few synaptophysin-immunoreactive profiles abutting GnRH neuron dendrites and cell bodies met the criteria for a synapse (e.g. *Figure 1*.C(ii) and F(i) do not). The density of synaptophysin-immunoreactive elements forming synaptic appositions along the first 60 μm (252 μm post-expansion) of the primary dendrite (soma-dendritic zone) was 0.8 ± 0.3 synapses/10 μm membrane with 95% of dendrites (N = 4 female mice) exhibiting synapses (*Figure 1A–C*, *Table 1*). The same analysis at the level of the distal dendron found 2.0 ± 0.2 synapses/10 μm with 53% of dendritic elements having at least one synapse along the 15 μm (63 μm post-expansion) segment analysed (N = 4) (*Figure 1D–F*, *Table 1*). These observations indicate a 2.5-fold higher density of synaptic inputs onto GnRH neuron distal dendrons compared to their soma-dendritic zone suggesting a physiologically important role.

To understand the roles of electrical activity at these two dendritic domains in regulating GnRH secretion, a series of targeted inhibitory chemogenetic and optogenetic studies were undertaken in ovariectomized (OVX) female mice that display robust pulsatile LH secretion, and in OVX, estradiol + progesterone-treated (OVX+E+P) female mice that exhibit a time-locked LH surge at the time of lights off (*Figure 2—figure supplement 1A–C*). As the GnRH neurons elaborate very long projections within the hypothalamus, it is possible to selectively modulate their proximal soma-dendritic domains within the rostral preoptic area (rPOA) or their distal dendrons lying adjacent to the median eminence (ME) within the mediobasal hypothalamus (MBH)(*Figure 1*).

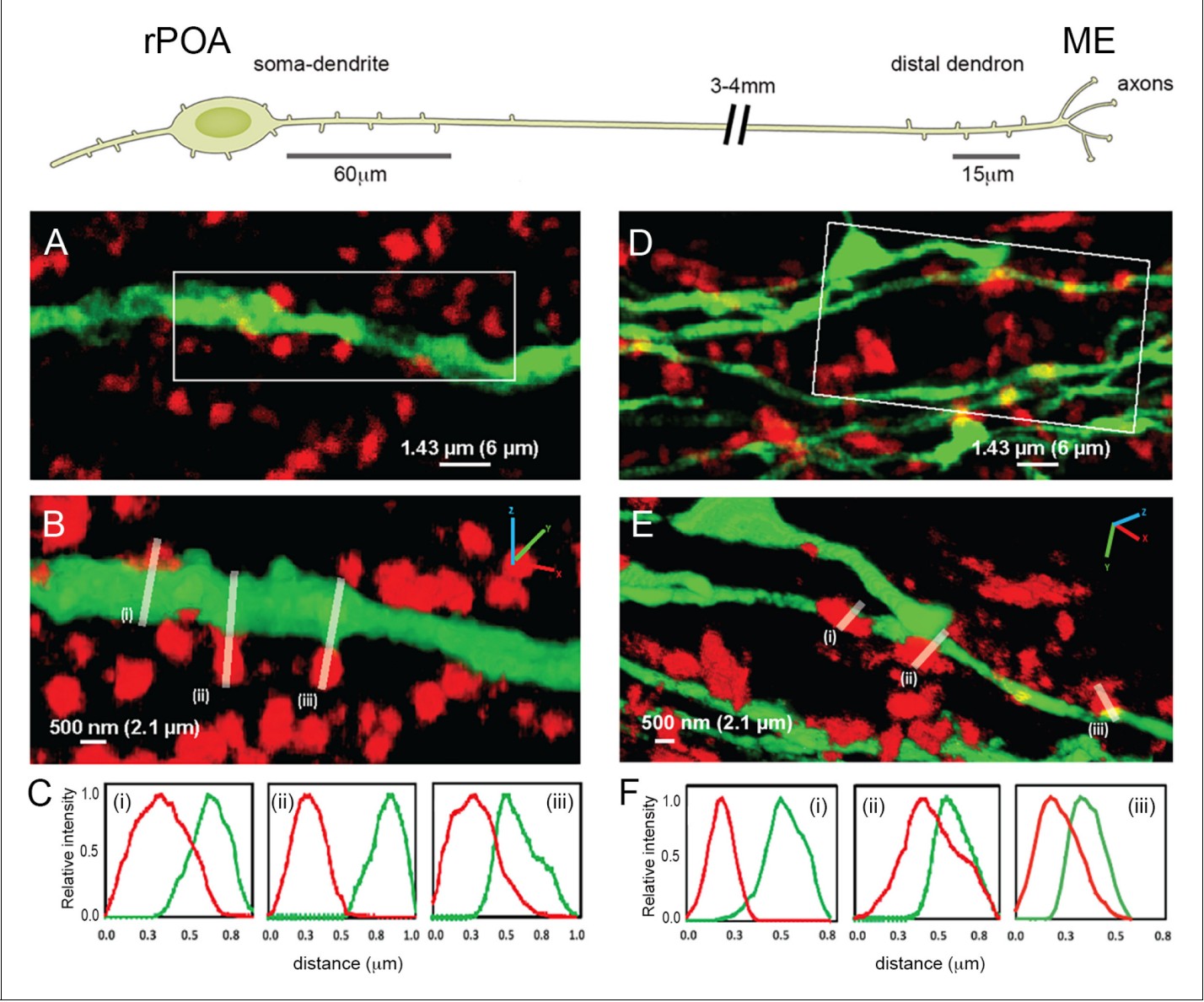

**Figure 1.** Expansion microscopy views of synaptic appositions at the GnRH neuron proximal dendrite and distal dendron. Schematic showing the morphology of a hypophysiotropic GnRH neuron with its soma-proximal dendrites located in the rostral preoptic area (rPOA) and the distal dendron and short axon branches in the median eminence (ME). Synaptic density analysis was undertaken on 60 µm-lengths of proximal dendrite and 15 µm-lengths of distal dendron. (A) Expansion microscopy view of a proximal dendrite (green) with surrounding synaptophysin puncta (red). (B) shows rotated 3D reconstruction with white lines indicating three appositions that were examined. (C) The side-on relative fluorescence intensity profiles are shown for the three appositions. (i) and (iii) represent synaptic appositions whereas (ii) indicates apposing profiles with no overlap that do not represent a synapse. (D) Expansion microscopy view of distal dendrons (green) with surrounding red synaptophysin puncta. (E) shows rotated 3D reconstruction with white lines indicating three appositions that were examined. (F) The relative fluorescence intensity profiles are shown for the three appositions. (ii) and (iii) represent synaptic appositions, whereas (i) indicates apposing profiles with no overlap that do not represent a synapse. Scale bars show pre-expansion values with expanded size in brackets. X-axis plots show pre- expansion values.

The online version of this article includes the following figure supplement(s) for figure 1:

**Figure supplement 1.** Definition of GABA$_A$ receptor synapses on GnRH neurons using ExM.

## Global suppression of GnRH neuron activity abolishes the LH surge and dampens pulsatile LH secretion

In the first series of experiments, we sought to establish the effects on LH secretion of suppressing activity within all compartments of the hypophysiotropic GnRH neurons projecting to the ME. GnRH-

**Table 1.** Quantitative analysis of synaptic appositions on the proximal and distal dendrites of GnRH neurons in diestrus female mice (N = 4).

For proximal dendrites, 60 μm of dendrite was measure in each case, whereas 15 μm lengths were examined for distal dendrites.

| GnRH proximal dendrites | |
| --- | --- |
| No. of dendrites counted | 38 |
| Percentage of dendrites with synaptophysin contact | 95% |
| No. of synaptophysin contacts with GnRH dendrite | 183 |
| Synaptic density per 10 μm | 0.8 ± 0.3 |
| **GnRH Distal Dendrites** | |
| No. of dendrites counted | 211 |
| Percentage of dendrites with synaptophysin contact | 53% |
| No. of synaptophysin contacts with GnRH dendrite | 330 |
| Synaptic density per 10 μm | 2.0 ± 0.2 |

Cre and control wild-type mice were injected with AAV2-Retro-hSyn-DIO-hM4D(Gi)-mCherry into the ME to enable retrograde targeting of hM4Di to the hypophysiotropic GnRH neuron population (*Figure 2A*; *Campos and Herbison, 2014*). Intraperitoneal or central administration of CNO selectively activates hM4D(Gi) receptors resulting in the activation in inward-rectifying potassium channels that suppress electrical excitability (*Roth, 2016*; *Smith et al., 2016*). Cell-attached recordings from mCherry-tagged GnRH neurons in acute brain slices prepared from AAV-injected GnRH-cre mice demonstrated that CNO significantly reduced the firing of GnRH neurons (*Figure 2—figure supplement 2*). Dual label immunohistochemistry showed that 64 ± 2.6% of rPOA GnRH neurons expressed hM4Di-mCherry (*Figure 2B* and *Figure 2—figure supplement 1D*) (N = 6). Prior estimates indicate that ~70% of rPOA GnRH neurons are hypophysiotropic (*Campos and Herbison, 2014*; *Merchenthaler et al., 1989*). A small number of cells (9 ± 1.2%) immunoreactive for mCherry but not GFP were detected and presumably represent GnRH neurons with low expression of GFP. No mCherry was detected in wild-type mice. Initial studies using hourly tail-tip bleeding in OVX+E+P wild-type and GnRH-Cre mice that had not received AAV injections demonstrated that a LH surge commenced between 17:00 and 18:00 hr (*Figure 2—figure supplement 1A–C*). As such, subsequent LH surge studies employed four blood samples taken at 16:00 (baseline) and at 19:00, 20:00 and 21:00 hr in each mouse to test for the presence of an LH surge.

In the first experiment, AAV-injected OVX+E+P wild-type and GnRH-Cre mice were given intraperitoneal saline or CNO at 16:00 hr. Whereas an LH surge occurred in all saline- (N = 9) and CNO- (N = 11) treated wild-type mice (*Figure 2C,D,G*), as well as saline-treated GnRH-Cre mice (N = 12) (*Figure 2E,G*), the LH surge was abolished in CNO-treated GnRH-Cre mice (N = 11)(*Figure 2F,G*). Mean LH levels in all three surging groups (*Figure 2G*) were the same as those observed in mice that had not received AAV injections (*Figure 2—figure supplement 1C*), while LH levels were significantly reduced at both 19:00 (p<0.001, two-way repeated ANOVA) and 20:00 time points in CNO-treated GnRH-Cre mice (p<0.05, two-way repeated ANOVA) (*Figure 2G*).

To next assess the effects of global GnRH neuron inhibition on pulsatile LH secretion, the same experimental strategy was undertaken in AAV-injected OVX GnRH-Cre mice. Three-minute blood samples were taken from the tail tip of mice for 2 hr with saline or CNO given 1 hr beforehand. Mice treated with saline exhibited a normal pulsatile pattern of LH release with ~3 ng/ml amplitude pulses occurring every ~15 min (N = 6)(*Figure 2H,I,L,M*). In contrast, mice treated with CNO exhibited much reduced LH levels with significantly fewer and small amplitude (~0.5 ng/ml) LH pulses (N = 6) (p<0.001,Mann-Whitney test)(*Figure 2J–M*). Mean levels of LH were 2.7 ng/ml in saline and 0.8 ng/ml (p<0.001, Mann-Whitney test) in CNO-treated mice (*Figure 3—figure supplement 1B,C*). Together, these studies demonstrate that the chemogenetic inhibition of the entire hypophysiotropic GnRH neuron prevents the occurrence of the LH surge and substantially reduces pulsatile secretion.

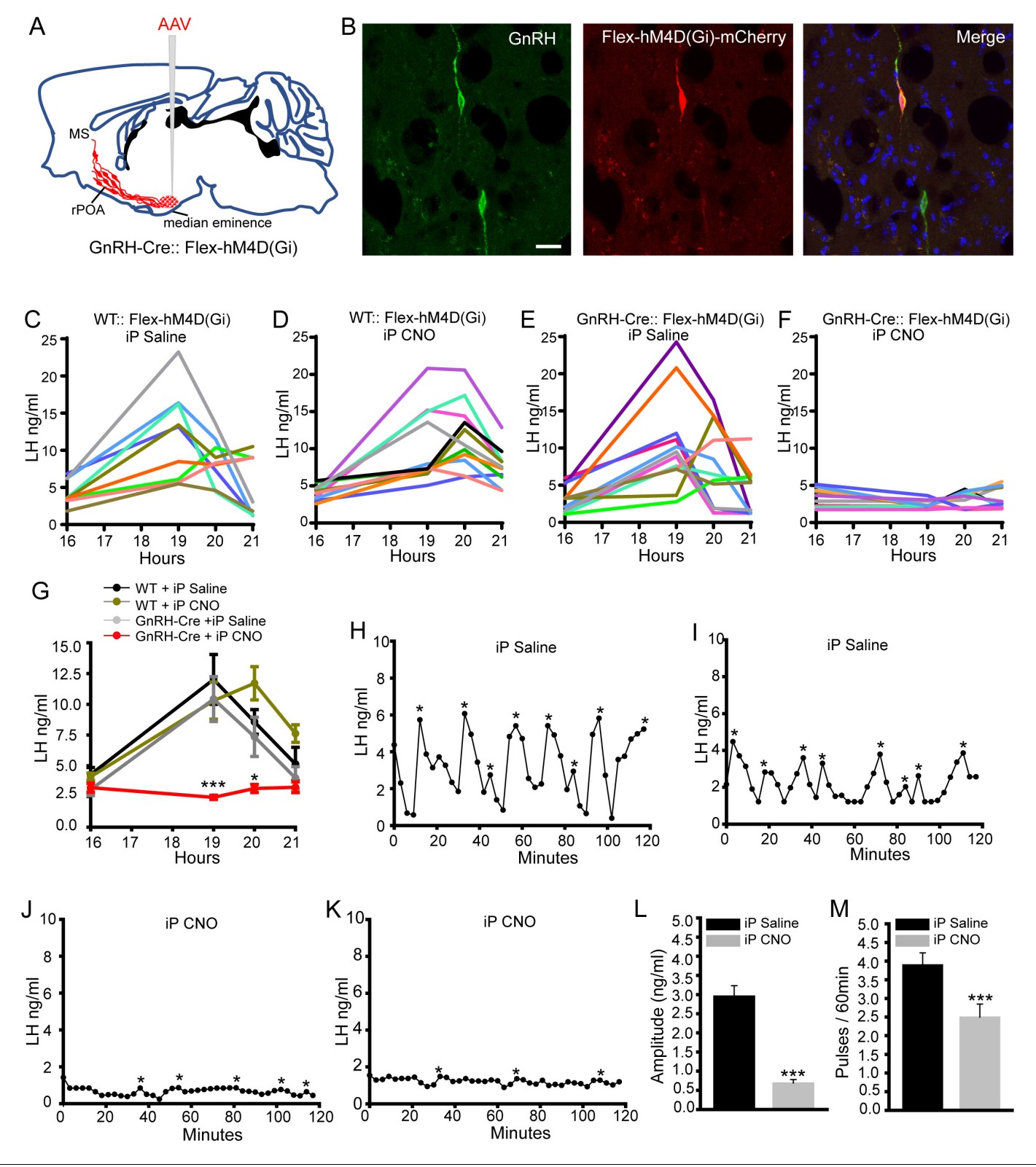

**Figure 2.** Chemogenetic inhibition of GnRH neurons by intraperitoneal (iP) injection of CNO suppresses both the surge and pulse profiles of LH secretion. (A) Schematic showing experimental protocol with GnRH-Cre mice injected with Cre-dependent Flex-hM4D(Gi)-mCherry AAVs bilaterally into the region of the median eminence and CNO given by IP injection. MS, medial septum. CNO, Clozapine N-oxide. (B) Fluorescence images of GnRH neurons expressing GnRH (green) and mCherry (red) in GnRH::Flex-hM4D(Gi)-mCherry mice. Scale bar, 20 μm. (C,D) LH profiles for all of the wild-type

*Figure 2 continued on next page*

Figure 2 continued

(WT) OVX+E+P mice given saline control (C, n = 9) or CNO (D, n = 11). (E,F) LH profiles for all of the GnRH-Cre OVX+E+P mice given saline control (E, n = 12) or CNO (F, n = 11). (G) Mean (± SEM) LH levels for the four experimental groups. *p<0.05, ***p<0.001, two-way repeated measures ANOVA with Holm-Sidak test. (H-M) Representative examples of pulsatile LH secretion in OVX GnRH-Cre mice given IP saline (H,I) or CNO (J,K). LH pulses are indicated by asterisks. (L,M) Mean (± SEM) amplitude and frequency of LH pulses in saline (n = 6) and CNO (n = 6). ***p<0.001 Mann-Whitney U-tests. The online version of this article includes the following figure supplement(s) for figure 2:

**Figure supplement 1.** LH surge profiles of wild-type and un-injected GnRH-Cre mice and expression of hM4D(Gi)-mCherry in GnRH neurons.
**Figure supplement 2.** Inhibition of GnRH neurons expressing hM4D(Gi) by CNO.

## Suppression of excitability at GnRH neuron distal dendrons and axons abolishes the LH surge and dampens pulsatile LH secretion

In these experiments, GnRH-Cre mice were injected with AAV2-Retro-hSyn-DIO-hM4D(Gi)-mCherry in exactly the same manner as above but CNO was injected directly into the MBH (*Figure 3A*) to assess the importance of GnRH neuron distal dendron and axon excitability in the different LH secretion modes. Following AAV retrograde transport to the cell body, hM4Di-mCherry is expressed throughout the GnRH neuron including the distal dendron and axons in the ME (*Figure 3B*). Injection of saline into the MBH of AAV-injected OVX+E+P GnRH-Cre mice (N = 12) had no effect on the occurrence of the LH surge (*Figure 3C,E*), whereas injection of CNO (N = 12) abolished the LH surge in all mice (*Figure 3D,E*). LH levels were significantly reduced at 19:00, 20:00 and 21:00 time points in CNO-treated GnRH-Cre mice (p<0.001, two-way repeated ANOVA) (*Figure 3E*).

The same strategy employed in AAV-injected OVX GnRH-Cre mice resulted in normal LH pulsatility in saline-treated animals (N = 8)(*Figure 3F,G*), but a substantial reduction in both the amplitude (p<0.001, Mann-Whitney test) and frequency (p<0.05, Mann-Whitney test) of LH pulses in mice receiving CNO (N = 7)(*Figure 3H–K*). Mean levels of LH were 3.2 ng/ml in saline and 0.7 ng/ml (p<0.001, Mann-Whitney test) in CNO-treated mice (*Figure 3—figure supplement 1D,E*). These results suggest that action potential propagation and/or synaptic integration within the distal compartment of hypophysiotropic GnRH neurons is obligatory for the LH surge and strongly facilitatory for pulsatile LH secretion.

## Suppression of excitability at the GnRH neuron soma-dendritic zone abolishes the LH surge but has no effect upon pulsatile LH release

The next experiments were undertaken to assess the importance of the GnRH neuron soma-proximal dendritic zone in pulse and surge secretion. GnRH-Cre mice were injected with AAV2-Retro-hSyn-DIO-hM4D(Gi)-mCherry into the ME as above and a large volume (2 µL) of saline or CNO injected into the rPOA at 16:00 hr (*Figure 4A*). Although an LH surge occurred in all saline-treated mice (N = 9)(*Figure 4B*), this was abolished in mice receiving CNO into the rPOA (N = 10)(*Figure 4C,D*). The levels of LH were significantly reduced at 19:00, 20:00 and 21:00 time points in CNO-treated GnRH-Cre mice (p<0.001, two-way repeated ANOVA) (*Figure 4D*).

For pulsatile LH secretion, the same injection of saline (N = 8) or CNO (N = 9) into the rPOA of AAV-injected OVX mice 10 min prior to blood sampling was found to have no effect upon pulsatile LH secretion with no differences detected in LH profiles (*Figure 4E–H*), mean pulse amplitude (*Figure 4I*), frequency (*Figure 4J*) or mean LH levels (*Figure 3—figure supplement 1F,G*). This indicates that the functional integrity of the GnRH neuron soma-dendritic zone is critical for the generation of the GnRH/LH surge but that this is not the case for pulsatile secretion.

To confirm this surprising observation, we repeated the experiment but used an inhibitory optogenetic approach to suppress electrical activity at the GnRH neuron soma-dendritic zone (*Figure 5A*). Archaerhodopsin-3 was expressed selectively in GnRH neurons by crossing GnRH-Cre and FLEX-Rosa-CAG-Arch-GFP[+/+] mouse lines resulting in 96±2% GnRH cell bodies expressing GFP (*Figure 5B*, *Figure 5—figure supplement 1A*) with extension into the long projections to the ME. Brain slice electrophysiological studies in these mice demonstrated that short periods (1–10 s) of yellow light (593 nm) illumination hyperpolarized and effectively silenced GnRH neuron firing (*Figure 5—figure supplement 1B,C*). Continuous illumination for periods of minutes also significantly reduced firing but showed variable efficacy (*Figure 5—figure supplement 1E,F*), while a 10 s on 10

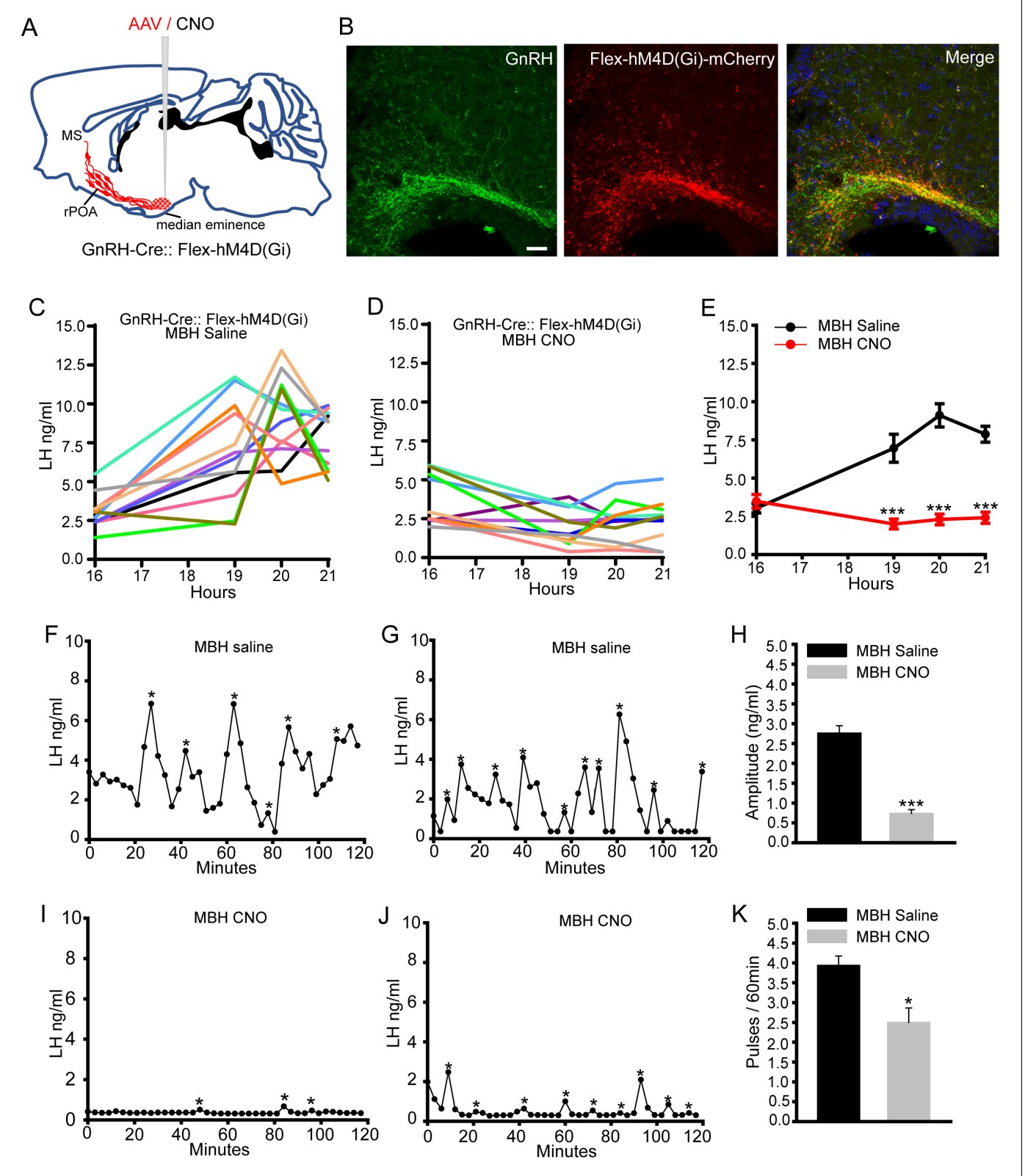

**Figure 3.** Chemogenetic inhibition of GnRH neuron distal dendron activity suppresses both the surge and pulse profiles of LH secretion. (**A**) Schematic showing experimental protocol with GnRH-Cre mice injected with Cre-dependent Flex-hM4D(Gi)-mCherry AAVs bilaterally into the region of the median eminence and CNO given into the same region. MS, medial septum. CNO, Clozapine N-oxide. (**B**) Fluorescence images of GnRH neuron projections in the vicinity of the median eminence expressing GnRH (green) and mCherry (red) in GnRH::Flex-hM4D(Gi)-mCherry mice. Scale bar, 20

*Figure 3 continued on next page*

*Figure 3 continued*

μm. (C,D) LH surge profiles in all GnRH-Cre OVX+E+P female mice following MBH injection of saline (C, n = 12) or CNO (D, n = 12). (E) Mean (± SEM) LH levels. ***p<0.001, two-way repeated measures ANOVA with Holm-Sidak test. F-J. Representative examples of pulsatile LH secretion in OVX GnRH-Cre mice given saline (F,G) or CNO (I,J). LH pulses are indicated by asterisks. Mean (± SEM) amplitude (H) and frequency (K) of LH pulses in saline (n = 8) and CNO (n = 7). ***p<0.001,*p<0.05 Mann-Whitney U-tests.

The online version of this article includes the following figure supplement(s) for figure 3:

**Figure supplement 1.** Photomicrographs showing location of cannula and mean (± SEM) LH levels in GnRH-Cre OVX mice with saline and CNO.

s off pattern of light illumination remained effective in suppressing firing during illumination for minutes (*Figure 5—figure supplement 1D*).

The LH surge occurs with the same profile of wild-type mice in GnRH-Cre[+/-];FLEX-Rosa-CAG-Arch-GFP[+/+] mice (*Figure 5—figure supplement 2A–B*). As this event is prolonged, we used an intermittent 593 nm light illumination strategy over 4.5 hr bilaterally into the rPOA to control the activity of the GnRH neurons between 16:00 and 20:30 hr (*Figure 5D*). This consisted of a '30 min on, 30 min off' protocol with 10 Hz light given '10 s on, 10 s off' predicted (*Figure 5—figure supplement 1D*) to stop electrical activity every alternate 10 s during each '30 min on' period. Control experiments showed that i) implanted OVX+E+P wild-type mice given this illumination strategy exhibited a normal LH surge (*Figure 5—figure supplement 2C–E*), ii) implanted OVX+E+P GnRH-Cre[+/-];FLEX-Rosa-CAG-Arch-GFP[+/+] mice without laser illumination exhibited normal LH surges (N = 11) (*Figure 5C*) and iii) implanted OVX+E+P GnRH-Cre[+/-];FLEX-Rosa-CAG-Arch-GFP[+/+] mice given incorrect 473 nm blue light illumination also had normal LH surges (*Figure 5E,F*). In contrast, the LH surge was abolished completely with bilateral yellow light illumination of the rPOA (N = 12) (*Figure 5D,F*).

As LH pulses in this OVX model occur approximately every 15 min, we reasoned that a continuous 10 Hz light illumination of the rPOA for 40 min would be sufficient to evaluate the effect of suppressing soma-proximal dendrite activity on LH pulsatility. Control experiments demonstrated that this laser illumination strategy did not alter pulsatile LH secretion in wild-type mice (*Figure 5—figure supplement 2F,I*) with pulse amplitude (*Figure 5—figure supplement 2G,J*), frequency (*Figure 5—figure supplement 2H,K*) and mean LH levels (*Figure 5—figure supplement 2L,M*) all remaining unchanged. Three-minute tail tip blood samples were taken from OVX GnRH-Cre[+/-];FLEX-Rosa-CAG-GFP[+/+] mice for 2 hr with bilateral 593 nm yellow light illumination of the rPOA occurring from 40 to 80 min. This was found to have no effect on pulsatile LH secretion (*Figure 5G,H*) with pulse amplitude (*Figure 5I*), frequency (*Figure 5J*) and mean LH levels (*Figure 5O*) remaining unchanged during the illumination period. As a control, the same mice were also illuminated with 473 nm blue light with no effects on pulsatile LH secretion being detected (*Figure 5K–N,P*). These observations parallel those of the POA chemogenetic inhibition strategy with optogenetic suppression of soma-proximal dendritic excitability abolishing the LH surge but having no effect upon LH pulsatility.

## Discussion

It is well established that dendritic computations and integration across both local and global scales are critical for generating precise patterns of neural activity (*Stuart and Spruston, 2015*). We demonstrate here an unusual scenario in which the GnRH neuron utilizes anatomically distinct dendritic compartments to generate its two different modes of secretion: the pulse and surge. Whereas global or local distal dendron inhibition of hypophysiotropic GnRH neurons prevents the surge and greatly reduces pulsatile secretion, local soma-proximal dendrite inhibition abolishes the surge but has no effect on pulses. This indicates that the soma and proximal dendrites of the GnRH neuron are critical for the occurrence of the GnRH surge, whereas the distal dendritic zones are sufficient on their own for generating pulsatile release.

The use of chemogenetic and optogenetic inhibitory approaches require careful controls due, primarily, to the possibility of off-target effects of clozapine derived from CNO (*Gomez et al., 2017*) and the potential warming of brain tissue by lasers (*Owen et al., 2019*). We have previously reported that CNO administered to wild-type mice has no impact upon pulsatile LH secretion (*Coutinho et al., 2020*) and demonstrate here that, equally, CNO has no impact on the LH surge in

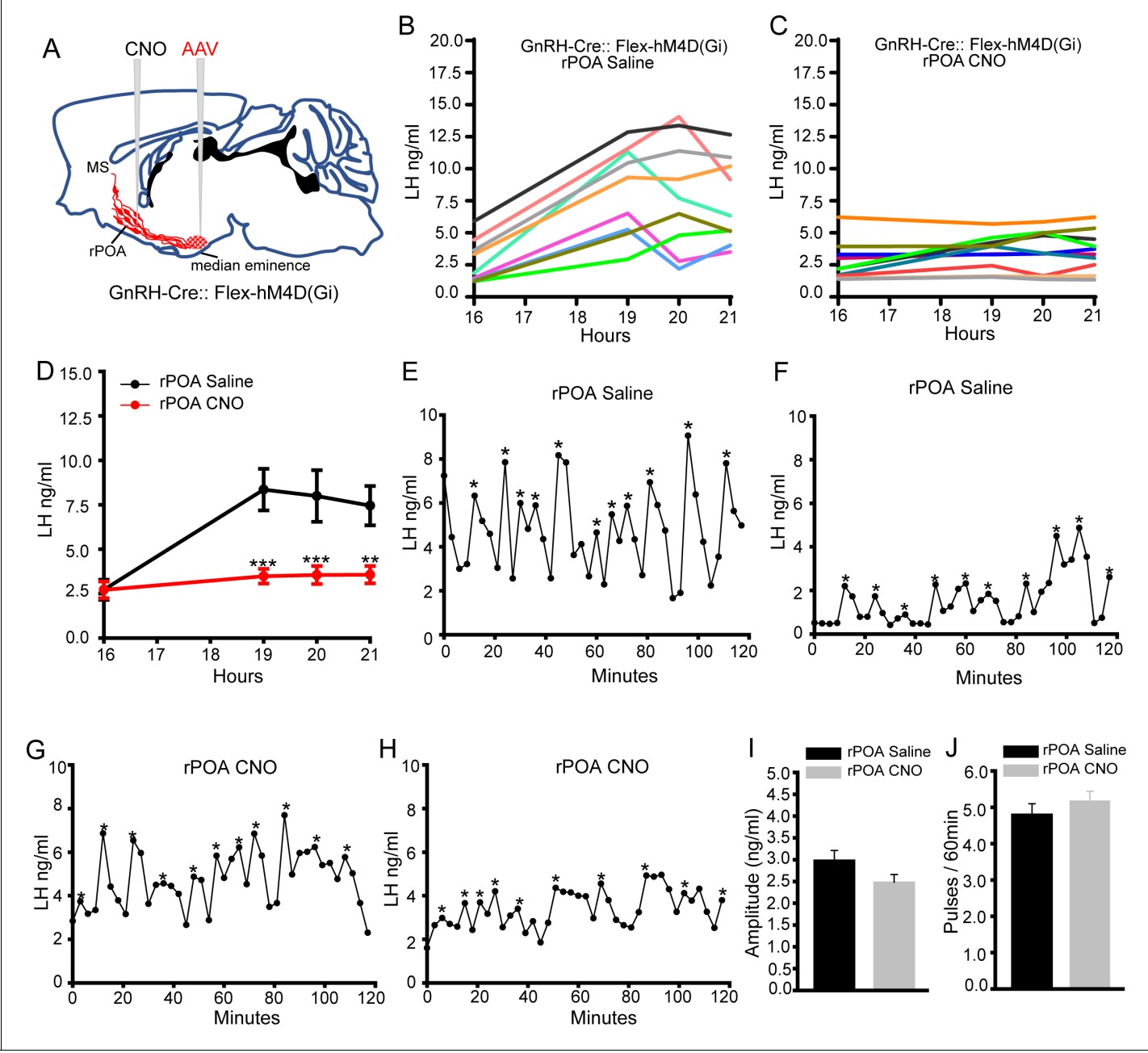

**Figure 4.** Chemogenetic inhibition of GnRH neuron soma-proximal dendrite activity suppresses only the LH surge. (**A**) Schematic showing experimental protocol in which GnRH-Cre mice are injected with Cre-dependent Flex-hM4D(Gi)-mCherry AAVs bilaterally into the region of the median eminence and, later, CNO injected into the rostral preoptic area (rPOA). MS, medial septum. CNO, Clozapine N-oxide. (**B,C**) LH surge profiles in all GnRH-Cre OVX+E+P female mice following rPOA injection of saline (**C**, n = 9) or CNO (**D**, n = 10). (**D**) Mean (± SEM) LH levels. **p<0.01, ***p<0.001, two-way repeated measures ANOVA with Holm-Sidak test. (**E-H**), Representative profiles of LH pulsatile secretion in GnRH-Cre OVX female mice with rPOA injection of saline (**E, F**) or CNO (**G, H**). LH pulses are indicated by asterisks. Mean (± SEM) LH pulse amplitude (**I**) and frequency (**J**) in GnRH-cre OVX mice with rPOA injection of saline (n = 8) and CNO (n = 9).

wild-type mice. This indicates that CNO and its derivatives are unable to modulate the GnRH neuronal network in the absence of a CNO-activated receptor. We have also performed both yellow and blue light laser controls in these studies and show that the activating yellow light has no impact on LH pulses or the LH surge in wild-type mice and that switching to the incorrect wavelength blue light in Arch3-expressing mice is similarly without impact on either mode of secretion.

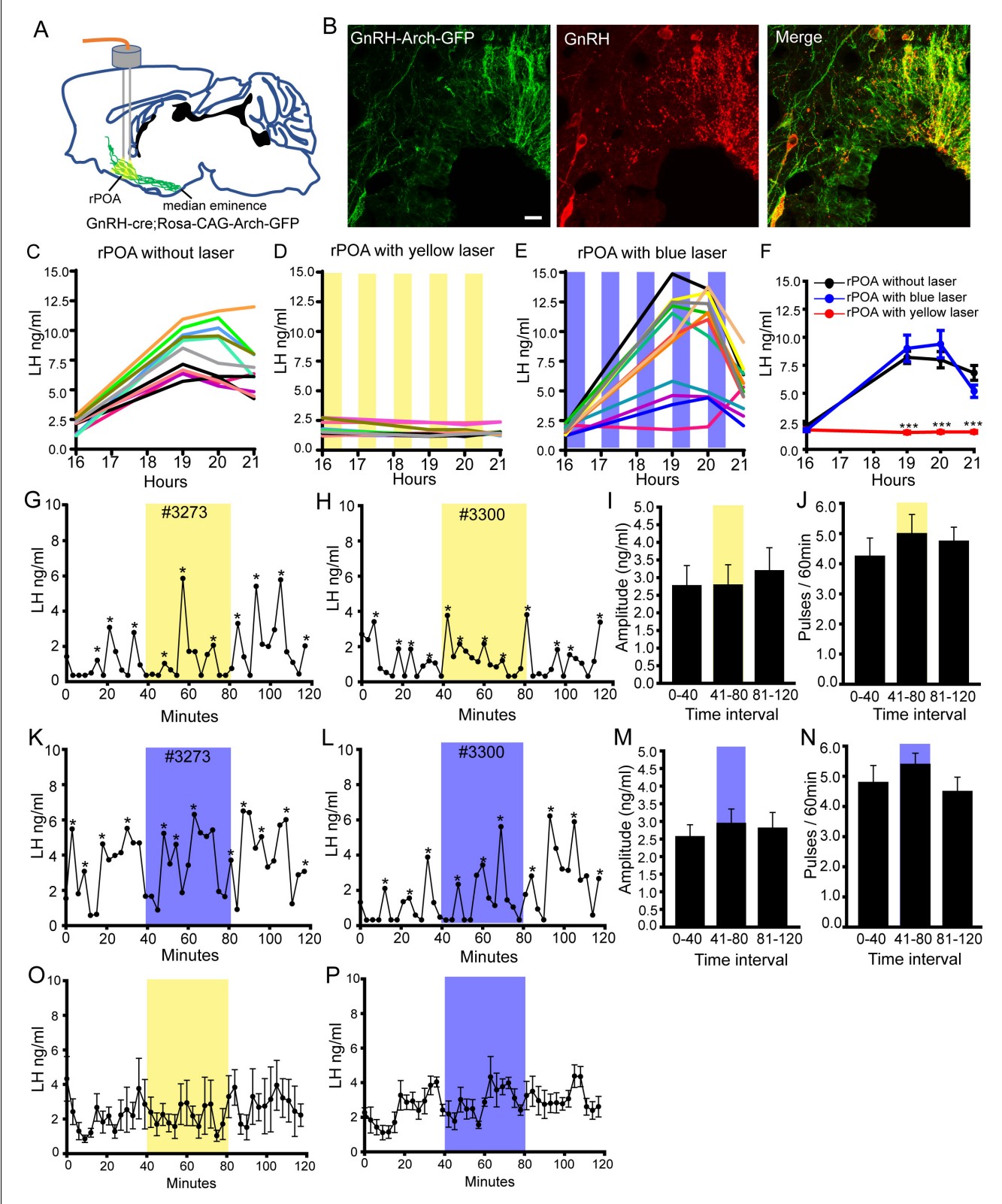

**Figure 5.** Bilateral optogenetic inhibition of GnRH neuron soma-proximal dendrite activity suppresses only the LH surge. (A) Schematic showing experimental protocol with GnRH-cre[+/-];Rosa-CAG-Arch-GFP[+/+] mice with bilateral optic fiber placement in the rostral preoptic area (rPOA). (B) Fluorescence images of GnRH neurons expressing Archaerhodopsin (GFP reporter) and GnRH (red) in GnRH-cre[+/-];Rosa-CAG-Arch3-GFP[+/+] mice. Scale bar, 20 μm. (C-E) LH surge profiles in all control GnRH-cre[+/-];Rosa-CAG-Arch-GFP[+/+] OVX+E+P female mice (C, n = 11) and those given intermittent

*Figure 5 continued on next page*

*Figure 5 continued*

(yellow shaded areas) bilateral rPOA 593 nm illumination at 10 Hz (D, n = 12) or, as a control, 473 nm (E, n = 12). (F) Mean (± SEM) LH levels. ***p<0.001, two-way repeated measures ANOVA with Holm-Sidak test. (G-N) Representative profiles of pulsatile LH secretion in GnRH-cre$^{+/-}$;Rosa-CAG-Arch3-GFP$^{+/+}$ OVX female mice given 40 min (shaded areas) bilateral rPOA illumination at 593 nm (G, H) or, as a control, 473 nm (K, L). LH pulses are indicated by asterisks. Mean (± SEM) LH pulse amplitude and frequency are given for yellow (I,J, n = 7) and blue (M,N, n = 6) light illumination. (O, P) Mean (± SEM) LH levels in GnRH-cre$^{+/-}$;Rosa-CAG-Arch-GFP$^{+/+}$ OVX mice with bilateral rPOA laser illumination at 593 nm (O, n = 7) or 473 nm (P, n = 6).

The online version of this article includes the following figure supplement(s) for figure 5:

**Figure supplement 1.** Archaerhodopsin expression in GnRH neurons and effects of yellow light on GnRH neuron firing.

**Figure supplement 2.** LH surge in GnRH-Arch mice and optogenetic control experiments showing effects of yellow light on the LH surge and pulses in wild-type mice.

Substantial evidence indicates that the GnRH surge arises from direct synaptic activation of the soma-proximal dendritic region of rPOA GnRH neurons. The majority of hypophysiotropic GnRH neurons in the rPOA express cFos at the time of the GnRH/LH surge (*Hoffman et al., 1993*) and evidence suggests that neighboring kisspeptin neurons provide a key stimulatory synaptic input to the GnRH neuron at this time (for review see *Herbison, 2016*; *Moenter et al., 2020*). This concept is supported here by the observation that both chemogenetic and optogenetic inhibition of GnRH neurons within the rPOA abolishes the LH surge. It is notable that the optogenetic activation of archaerhodopsin-3 does not completely suppress firing but entrains a 10 s on 10 s off pattern of firing every 30 min. As this was effective in stopping the surge, this suggests that the overall pattern of GnRH neuron firing may be more important than any average firing rate for surge induction. We also note that the chemogenetic inhibition of the distal GnRH projections was effective in blocking the LH surge indicating that action potential transmission to the GnRH terminals and/or further synaptic integration at the distal dendritic zone is necessary for the surge.

Accruing evidence has supported the possibility that the long, blended dendrite/axon (dendron) projections of GnRH neurons may represent an autonomous regulatory centre close to the GnRH secretory zone in the median eminence. Prior electron microscopic investigations have revealed the presence of synapses on these distal dendrons (*Moore et al., 2018b*) and, using expansion microscopy, we show here that the density of synaptic inputs at the distal dendrons is double that of the proximal dendrite. Thus, the distal dendron appears to be the most densely innervated dendritic compartment of the GnRH neuron. Studies in the brain slice indicate that GABA, glutamate and kisspeptin are all active in regulating the activity of the GnRH neuron distal dendron (*Herde et al., 2013*; *Iremonger et al., 2017*). Furthermore, it seems very likely that the arcuate nucleus (ARN) kisspeptin neurons target this dendritic zone to generate pulsatile GnRH/LH secretion (*Herbison, 2018*; *Plant, 2019*).

The inhibition strategies used in this study are not able to determine whether changes in GnRH secretion arise from alterations in intrinsic excitability or synaptic integration in the GnRH neuron; both will ultimately be inhibited by hyperpolarization of the dendron. However, it seems very likely that the suppression of episodic GnRH secretion results from alterations in synaptic integration; first, there is very little evidence that the intrinsic properties of adult GnRH neurons contribute to the generation of LH pulses (*Herbison, 2018*). Second, we show here that there is substantial synaptic density at the dendron and third, there is robust evidence in several species for the kisspeptin neuron pulse generator to operate synaptically to drive LH pulses (*Herbison, 2018*; *Moore et al., 2018a*; *Plant, 2019*).

We find here that the local chemogenetic inhibition of distal GnRH neuron projections in and around the median eminence exerts a very substantial suppression of pulsatile LH release but it does not stop entirely. Notably, the same result was observed with chemogenetic inhibition of the entire GnRH neuron. This may result from the CNO activation of G$_i$ signalling within GnRH distal projections being insufficient to arrest completely the potent G$_{q/11}$-mediated effects of kisspeptin on both TRPC and voltage-gated calcium channels in the dendron (*Iremonger et al., 2017*; *Smith et al., 2016*).

Neither chemogenetic nor optogenetic inhibition of rPOA GnRH neuron soma and dendrites had any effect on pulsatile LH secretion despite abolishing the LH surge. This demonstrates that the soma and proximal dendritic compartments of rPOA GnRH neurons are not required for the

generation of LH pulses. As the action potential initiation zones for GnRH neurons are only located in these proximal regions (*Herde and Herbison, 2015*; *Herde et al., 2013*), this suggests that the propagation of action potentials down the dendron are not required for pulsatile secretion. This is compatible with the demonstration of episodic GnRH secretion in in vitro preparations without GnRH cell bodies (*Purnelle et al., 1997*). Although we infused large volumes of CNO into the rPOA, that would have been expected to infiltrate the entire region, it remains that hypophysiotropic GnRH neurons located in the dorsal aspects of the medial septum and caudal-most in the MBH may have been unaffected. It is possible that these few GnRH neurons are not necessary for surge generation but are, nevertheless, able to maintain pulsatile LH secretion. However, this seems unlikely as reducing the number of hypophysiotropic GnRH neurons has a marked effect on LH pulse amplitude (*Czieselsky et al., 2016*) and we saw no change in LH pulse amplitude following either inhibition strategy.

It has recently become clear that the pulse generator is formed by ARN kisspeptin neurons that are intermittently synchronized and send excitatory projections to the GnRH neuron distal dendron in the mouse (*Clarkson et al., 2017*; *Herbison, 2018*; *Moore et al., 2018a*; *Plant, 2019*). This kisspeptin activation of the GnRH neuron distal dendrons and consequent GnRH secretion in the ME has repeatedly been shown to occur in the absence of action potential generation (*d'Anglemont de Tassigny et al., 2008*; *Glanowska and Moenter, 2015*; *Iremonger et al., 2017*). As such, it seems highly likely that the synaptic innervation of the distal dendron represents a highly localized, autonomous regulatory mechanism for the generation of pulsatile GnRH/LH secretion. The present observations provide the first in vivo evidence in support of this concept and demonstrate a novel mechanism for the synchronization of neural output within the forebrain. It is possible that this unusual situation arises from the unique embryonic migration of the GnRH neurons from the nose into the brain resulting in their somata being scattered throughout the basal forebrain but their projections being concentrated together just above the ME (*Wray, 2010*). The robust synaptic innervation of these clustered distal dendronic projections likely provides an efficient solution for the afferent synchronization of an otherwise widely dispersed neural population.

# Materials and methods

## Key resources table

| Reagent type (species) or resource | Designation | Source or reference | Identifiers | Additional information |
|---|---|---|---|---|
| Genetic reagent (*M. musculus*) | STOCK Tg(Gnrh1-cre)1Dlc/J | Jackson Laboratory | Stock #: 021207 RRID:IMSR_JAX:021207 | |
| Genetic reagent (*M. musculus*) | B6;129S-Gt(ROSA) 26Sor$^{tm35.1(CAG-AOP3/GFP)Hze}$/J | Jackson Laboratory | Stock #: 012735 RRID:IMSR_JAX:012735 | |
| Genetic reagent (*M. musculus*) | B6.DBA-Tg(Gnrh1-EGFP)1Phs | *Spergel et al., 1999* | MGI:6158458 | |
| Transfected construct (*M. musculus*) | AAV2-Retro-hSyn-DIO-hM4D(Gi)-mCherry | OBiO Technology (Shanghai) Corp., Ltd | RRID:Addgene_44362 | |
| Antibody | polyclonal guinea pig anti-GnRH antisera | gift G. Anderson, University of Otago | Cat #GA02 RRID:AB_2721118 | (1: 5000) |
| Antibody | polyclonal rabbit anti-GnRH antisera | gift G. Anderson, University of Otago | Cat #GA01 RRID:AB_2721114 | (1: 5000) |
| Antibody | Alexa Fluor 488-AffiniPure Donkey Anti-Guinea Pig IgG (H+L) | Jackson Immuno Research Labs | Cat# 706-545-148, RRID:AB_2340472 | (1:500) |
| Antibody | Alexa Fluor 594 donkey anti-rabbit antibody | Jackson Immuno Research Labs | Cat# 711-585-152, RRID:AB_2340621 | (1:500) |
| Antibody | DAPI (4',6-Diamidino-2-Phenylindole, Dilactate) antibody | Thermo Fisher Scientific | Cat# D3571, RRID:AB_2307445 | 300 nM |

*Continued on next page*

*Continued*

| Reagent type (species) or resource | Designation | Source or reference | Identifiers | Additional information |
|---|---|---|---|---|
| Antibody | Anti-GFP (chicken polyclonal) | Abcam | Cat# AB13970 RRID:AB_300798 | (1:8000) |
| Antibody | Anti-Synaptophysin 1 (guinea pig polyclonal) | Synaptic Systems | Cat#101004 RRID:AB_1210382 | (1:800) |
| Antibody | Anti-Vesicular GABA Transporter (rabbit polyclonal) | Synaptic Systems | Cat#131003 RRID:AB_887869 | (1:800) |
| Antibody | Anti- Gephyrin (guinea pig polyclonal) | Synaptic Systems | Cat#147318 RRID:AB_2661777 | (1:800) |
| Antibody | Goat anti-chicken (goat polyclonal, Alexa488-conjugate) | Thermo Fisher Scientific | Cat# A-11039 RRID:AB_2534096 | (1:200) |
| Antibody | Goat anti-guinea pig (goat polyclonal, biotin-conjugated) | Vector Laboratories | Cat# BA-7000 RRID:AB_2336132 | (1:200) |
| Antibody | Goat anti-rabbit ( goat polyclonal, ATTO647N-conjugated) | Sigma-Aldrich | Cat# 40839 RRID:AB_1137669 | (1:200) |
| Chemical compound, drug | Beta-Estradiol 3-benzoate | Sigma Aldrich | E8515 | |
| Chemical compound, drug | Beta estradiol | Sigma Aldrich | E8875 | |
| Chemical compound, drug | progesterone | Sigma Aldrich | P0130 | |
| Chemical compound, drug | clozapine-N- oxide | Sigma Aldrich | C0832 | |
| Software, algorithm | GraphPad Prism software | GraphPad Prism | RRID:SCR_002798 | Version 6.01 |
| Software, algorithm | ImageJ image analysis software | ImageJ (https://imagej.net/) | RRID:SCR_003070 | |
| Software, algorithm | Vaa3D data visualization software | Vaa3D (http://www.vaa3d.org) | RRID:SCR_002609 | |

## Animals

Adult littermate heterozygous *Gnrh1*-Cre$^{+/-}$ mice with Cre recombinase targeted to the first coding exon of *Gnrh1* (*Yoon et al., 2005*)(JAX #021207), alone or crossed onto Cre-dependent reporter ROSA26-CAG-Arch3-GFP lines (*Madisen et al., 2012*)(JAX #012735) (C57BL/6J genetic background), were housed under a 12:12 hr lighting schedule (lights on at 07:00-19:00) with ad libitum access to food and water. Genotypes were determined by PCR of mouse tail DNA samples. All animals were maintained and treated in accordance with the National Institutes of Health Guide for the Care and Use of Laboratory Animals and were approved by Animal Care and Use Committee of Shanghai JiaoTong University School of Medicine (2016–0016). Female C57BL/6 GnRH-GFP mice (*Spergel et al., 1999*) were housed under a 12:12 hr lighting schedule (lights on at 07:00-19:00) with ad libitum access to food and water at the University of Otago Hercus-Taieri Breeding Unit. All experiments were approved by the University of Otago Animal Welfare and Ethics Committee (96/17).

## Expansion microscopy synaptic density experiments

Diestrus GnRH-GFP mice were anaesthetized with pentobarbital and perfused transcardially with 4% paraformaldehyde in 0.1 M phosphate buffered saline (PBS; pH 7.6). Brains were post-fixed in the same fixative at room temperature for 1 hr and rinsed in 0.1 M PBS. Coronal brain sections (50 µm thick) were cut on a vibratome in Tris-buffered saline (TBS) and treated with 0.1% Triton-X-100% and 2% goat serum overnight at 4°C followed by 0.1% sodium borohydrate in TBS for 15 min at room

temperature, then washed and incubated for 72 hr at 4°C with well-characterized chicken anti-GFP (1:8,000; Abcam, AB13970) (*Dow et al., 2014*) and guinea-pig anti-synaptophysin 1 (1:800, Synaptic Systems, 101004) (*Wallrafen and Dresbach, 2018*) or rabbit anti-vesicular GABA transporter (VGAT, 1:800, Synaptic Systems, 131003) (*Martens et al., 2008*) and guinea pig anti-gephyrin (1:800, Synaptic Systems, 147318) (*Pan et al., 2019*) antisera in TBS containing 0.3% Triton-X-100, 0.25% bovine serum albumin and 2% goat serum. Sections were then washed and incubated with biotinylated rabbit anti-guinea pig immunoglobulins (1:200, Vector Laboratories), Alexa488-conjugated goat anti-chicken (1:200, ThermoFisher Scientific) and ATTO647N goat anti-rabbit immunoglobulins (Sigma-Aldrich) for 15 hr at 4°C. Omission of primary antibodies resulted in no detectable immunofluorescence. Sections were then expanded using a protocol similar to that published previously (*Chen et al., 2015*; *Chozinski et al., 2016*). In brief, trimmed immunostained sections underwent linking with anchoring agent (MA-NHS; 2 mM) for 1.5 hr before being incubated in monomer solution (1x PBS, 2 M NaCl, 8.625% (w/w) sodium acrylate, 2.5% acrylamide, 0.15% N,N′-methylenebisacrylamide) on ice for 45 min. Sections were then immersed in gelling solution (monomer solution added with 0.01% 4-hydroxy-2,2,6,6-tetramethylpiperidin-1-oxyl + 0.2% tetramethylethy lenediamine + 0.2% ammonium persulfate) and incubated in a humidified chamber at 37°C for 2 hr. Gel-embedded sections were trimmed and digested overnight with 8 U/mL Proteinase K in digestion buffer at 37°C and then rinsed in PBS and incubated with Strepavidin-568 at 37°C for 3 hr. Expansion was undertaken by adding water every 20 min, up to five times.

Images were acquired using a Nikon A1R upright confocal microscope equipped with a water-immersion lens (25x Numerical Aperture 1.1; Working Distance 2 mm) using sequential scanning mode and image stacks collected with 600 nm focus intervals. Sixteen-bit confocal images (1024 × 512 pixel format) were analyzed using ImageJ to determine the number of synaptophysin-immunoreactive boutons apposing GnRH neuron dendrites and the expansion factor. For the soma-dendritic zone, contiguous 250 μm (60 μm pre-expansion) lengths of primary dendrite arising from the GnRH cell body were selected at random from three rostral preoptic area sections in each of the four mice. Each apposing synaptophysin bouton (diameter >0.4 μm) was examined in three dimensions to establish the best side-on or face-on view of the imaged synapse. A line scan was then performed across this plane and the relative intensity of the Alexa488 and ATTO647 measured and plotted in Microsoft Excel. An apposition was considered a synapse where the signals overlapped by >0.95 μm (0.23 μm pre-expansion) in the side-on plane or by >1.75 μm (0.42 μm pre-expansion) in the face plane. The same process was used to establish the synaptic density for the distal dendron using 60 μm (15 μm pre-expansion) contiguous lengths of dendron located in the ventral arcuate nucleus immediately lateral to the margin of the median eminence (*Moore et al., 2018b*). Three-dimensional reconstruction of GnRH dendrites displaying synaptic inputs was undertaken with Vaa3D (*Peng et al., 2014*).

## Stereotaxic injections of AAVs and implantation of cannula or fibre optic

For the chemogenetic inhibition studies, recombinant cre-dependent AAVs encoding hM4Di (AAV2-Retro-hSyn-DIO-hM4D(Gi)-mCherry, $5 \times 10^{12}$ GC/mL; OBiO Technology (Shanghai) Corp., Ltd) were injected bilaterally into the region of the ME of GnRH-Cre$^{+/-}$ mice. Adult female mice (8–12 weeks) were anesthetized with 2% isoflurane, and placed in a stereotaxic apparatus. A Hamilton syringe apparatus holding a 10 μL Hamilton syringe with 25-gauge needle was used to perform bilateral injections. The stereotaxic coordinates were: AP −1.7 mm; ML ±0.3 mm; DV −5.8 mm. The needle was lowered into place and 1 μL AAV injected over 5 min, and the needle was left in situ for a further 5 min before being withdrawn. As appropriate, customized bilateral guide cannulae (RWD Life Science) were then implanted into the region of the ME (AP −1.7 mm; ML ±0.3 mm; DV −5.3 mm), or customized unilateral cannula guides (RWD Life Science) were implanted to the POA (AP 0.6 mm; ML ±0.0 mm; DV −4.5 mm).

For the optogenetic inhibition, female GnRH-Cre$^{+/-}$;Flex-Rosa-CAG-Arch3-GFP$^{+/+}$ mice aged 8–12 weeks were anesthetized with 2% isoflurane, placed in a stereotactic instrument and customized bilateral optic fibers (200 μm diameter, pitch 0.7 mm, 0.37 N.A.; Doric Lenses, Canada) implanted into the rPOA (AP 0.6 mm; ML ±0.35 mm; DV −4.5 mm) and secured with dental cement. All mice were given at least 14 days to recover before OVX surgery.

## LH pulse and surge models

At least 2 weeks after cranial surgery, female GnRH-Cre$^{+/-}$ and GnRH-Cre$^{+/-}$;Rosa-CAG-Arch3-GFP$^{+/+}$ mice were ovariectomized and 1 week later used for 3-min interval tail tip bleeding experiments from 9:00 AM to 11:00 AM to assess LH pulsatility (*Czieselsky et al., 2016*). To model the LH surge, AAV-injected and implanted mice were ovariectomized and given a s.c. SILASTIC capsule filled with 1 μg/20 g b.w. 17-β-estradiol that generates negative feedback (*Czieselsky et al., 2016*). Six days later, mice were given an injection of estradiol benzoate (1 μg/100 μL, s.c.) at 09:00. The following day, mice were given an injection of progesterone (500 μg/100 μL) at 09:00 and hourly tail-tip bleeding was performed from 16:00 to 21:00 in initial experiments and then at 16;00, 19:00, 20:00 and 21:00 in experimental mice. Lights went out at 19:00. Repetitive tail-tip pulse blood sampling was undertaken using the method of *Steyn et al., 2013*. In brief, after a single excision of the very tip of the tail, mice were gently restrained in a cardboard tube every 3 min while their tail was wiped clean with saline and then massaged so that a second investigator could take a 6-μL blood sample from the tail tip with a pipette. Whole blood was immediately diluted in 114 μL of 0.1M PBS with 0.05% Tween 20, vortexed, and snap frozen on dry ice. Samples were stored at −20°C for a subsequent LH ELISA (*Steyn et al., 2013*). LH pulses were identified as an increment in LH values from nadir to peak over 25% (peak minus nadir/nadir x100). Studies using GCaMP photometry assessments of GnRH pulse generator activity in short-term OVX mice show that every increment in LH >25% using this tail-tip procedure is correlated with a pulse generator event and represents a bone fide LH pulse (*Clarkson et al., 2017*; *McQuillan et al., 2019*).

## Chemogenetic inhibition experiments

Intraperitoneal injections of CNO (2 mg/kg [*Bock et al., 2013*], Sigma) or saline were given at 16:00, 3 hr before lights off in LH surge experiments, or 1 hr before blood sampling in LH pulse experiments. Intracranial injections of CNO or saline were given through bilateral or unilateral infusion cannulae (1 μL each side in ME or 2 μL in rPOA; 6 μM CNO [*Stachniak et al., 2014*]) at 16;00, 3 hr before lights off in LH surge experiment, or 10 min before beginning blood sampling in LH pulse experiments.

## Brain slice electrophysiology

GnRH-cre$^{+/-}$;Rosa-CAG-Arch-GFP$^{+/+}$ mice (P21-P35) were briefly anesthetized with isoflurane and brains dissected out and immediately immersed into ice-cold Gey's balanced salt solution (NaCl 130, KCl 4.9, CaCl$_2$ 1.5, MgSO$_4$ 0.3, MgCl$_2$ 11, KH$_2$PO$_4$ 0.23, Na$_2$HPO$_4$ 0.775, glucose 5, HEPES 25, NaHCO$_3$ 22 and NaOH 12 (in mM), pH = 7.2, oxygenated with 95% O$_2$/5% CO$_2$) for 3 min. The 300 μm-thick coronal slices were sectioned on vibratome (VT-1000s, Leica, Germany) in ice-cold Gey's balanced salt solution and transferred to a holding chamber filled with room-temperature artificial cerebrospinal fluid (ACSF, NaCl 124, KCl 3, CaCl$_2$ 2, MgCl$_2$ 1, NaH$_2$PO$_4$ 1.25, NaHCO$_3$ 26 and glucose 10 (in mM), pH = 7.3, oxygenated with 95% O$_2$ and 5% CO$_2$). After a recovery of 30 min, slices were transferred to a recording chamber constantly superfused with oxygenated ACSF and neurons recorded.

GnRH neurons were identified by GFP expression with a 40x objective under the blue light which was generated from the mercury lamp (C-HGFI, Nikon, Japan) and filtered by fluorescence cubes (470/20 nm, Zeiss Inc, Germany). Under a Zeiss upright DIC microscope (Examiner A1, Zeiss, Germany), neurons were patched with a glass pipette (5–10 MΩ) pulled by P-97 micropipette puller (Sutter Instruments, USA). The pipette solution contained the following (in mM): potassium gluconate 105, KCl 5, CaCl$_2$ 0.5, MgCl$_2$ 2, EGTA 5, HEPES 10, Mg-ATP 4, GTP-Na 0.5, sodium phosphocreatine 7, Lucifer Yellow 0.05%, pH 7.4 adjusted with KOH. After establishing the whole-cell configuration, 100 ms current was injected to probe the action potential threshold, and the optogenetic inhibition was tested at 25 pA above rheobase. Various patterns of 593 nm light pulses at ~5 mW/mm (Polygon-400, Mightex, USA) were delivered through the water immersion 40x objective to optically inhibit the Arch-expression neurons. Recordings were performed using MultiClamp 700B patch-clamp amplifier (Molecular Devices, USA), digitized by Digidata 1440 (Molecular Devices, USA), and analyzed by pClamp 10 (Molecular Devices, USA).

## Optogenetic inhibition experiments

GnRH-Cre$^{+/-}$;Rosa-CAG-Arch3-GFP$^{+/+}$ mice were attached to the laser patch cord in their home cages and left to acclimate for 15 min, and 3 min blood sampling (6 µL per sample) undertaken for 120 min as described above. For optogenetic inhibition in LH surge experiments, a '30 min on, 30 min off' protocol with 10 Hz 593 nm laser light (10 s on, 10 s off) was used from 16:00 to 20:30. In the LH pulse experiments, blood sampling was undertaken for 120 min with an initial 40 min followed by yellow light illumination for 40 min and a final 40 min period without light. The DPSS laser (593 nm, ~5 mW per fiber; Shanghai Dream Lasers Technology Co., Ltd.) was controlled by a stimulator (RIGOL Technologies, Inc). After 1 week, mice received a second experiment in which they received control blue light illumination (473 nm, ~5 mW per fiber; Shanghai Dream Lasers Technology Co., Ltd.) in the same manner as for yellow light.

## Immunofluorescence

Coronal brain sections covering the full length of the rPOA and MBH were used to determine the locations of cannula/fibre optic and/or processed for dual-label analysis. Mice were anesthetized with choral hydrate and perfused with saline followed by 4% paraformaldehyde in 0.1M PBS. The brains were quickly removed, post-fixed in 4% paraformaldehyde overnight, and subjected to dehydration in increasing saccharin solutions (20–30%) at 4℃. The frozen brain slices were sectioned at 25 µm on a vibratome (Leica). Brain sections were placed in 3% normal goat serum and 0.2% Triton-X for 1 hr and then incubated in polyclonal guinea pig or rabbit anti-GnRH antisera (1:5000, gift from Greg Anderson, University of Otago) overnight at 4℃. Slices were rinsed in PBS then incubated in donkey anti-guinea pig Alexa-488 antibody (Jackson Immunoresearch) in GnRH-cre$^{+/-}$::Flex-hM4D (Gi) mice or donkey anti-rabbit Alexa-594 antibody (Jackson Immunoresearch) in GnRH-cre$^{+/-}$;Rosa-CAG-Arch-GFP$^{+/+}$ mice for 1 hr and 4',6-diamidino-2-phenylindole (DAPI, Invitrogen) for 10 min, then mounted after rinsing. Dual fluorescence images were captured on an Olympus FV3000 confocal microscope.

## Statistical analysis

Statistical analyses were carried out with GraphPad Prism (GraphPad Software). When comparing the amplitude and frequency of LH pulse within two groups, data were compared using Mann-Whitney test. When comparing the amplitude and frequency of LH pulse within three groups, data were compared using parametric one-way ANOVA (with post hoc Tukey's multiple comparisons test). The firing rate and LH surge profiles were assessed using repeated-measures ANOVA. Differences were considered significant for $p < 0.05$. Values are given as mean ± SEM.

## Acknowledgements

The research was financially supported by the grants from the Natural Science Foundation of Shanghai (grant number: 18ZR1422600 to LW), National Key project (grant number: 2018YFC1003000 to YK) and National Natural Science Foundation of China (grant number: 81771533 to YK and 31400970 to LW), and the New Zealand Health Research Council (AEH) and Wellcome Trust (AEH).

## Additional information

### Funding

| Funder | Grant reference number | Author |
|---|---|---|
| Natural Science Foundation of Shanghai | 18ZR1422600 | Li Wang |
| National Natural Science Foundation of China | 81771533 | Yanping Kuang |
| National Key Research and Development Programs | 2018YFC1003000 | Yanping Kuang |
| National Natural Science Foundation of China | 31400970 | Li Wang |

| Health Research Council of New Zealand | 17-236 | Allan Edward Herbison |
| Wellcome Trust | | Allan Edward Herbison |

The funders had no role in study design, data collection and interpretation, or the decision to submit the work for publication.

### Author contributions

Li Wang, Data curation, Formal analysis, Supervision, Funding acquisition, Investigation, Writing - original draft, Writing - review and editing; Wenya Guo, Investigation, Methodology; Xi Shen, Software, Investigation, Methodology; Shel Yeo, Formal analysis, Investigation, Writing - review and editing; Hui Long, Software, Validation; Zhexuan Wang, Formal analysis, Investigation, Methodology; Qifeng Lyu, Conceptualization, Resources; Allan E Herbison, Conceptualization, Supervision, Funding acquisition, Writing - original draft, Writing - review and editing; Yanping Kuang, Conceptualization, Resources, Funding acquisition

### Author ORCIDs

Allan E Herbison  https://orcid.org/0000-0002-9615-3022

### Ethics

Animal experimentation: All animals were maintained and treated in accordance with the National Institutes of Health Guide for the Care and Use of Laboratory Animals and were approved by Animal Care and Use Committee of Shanghai JiaoTong University School of Medicine (2016-0016) and the University of Otago Animal Welfare and Ethics Committee (96/17).

### Decision letter and Author response

Decision letter https://doi.org/10.7554/eLife.53945.sa1
Author response https://doi.org/10.7554/eLife.53945.sa2

## Additional files

### Supplementary files

• Transparent reporting form

### Data availability

All data generated or analysed during this study are included in the manuscript and supporting files.

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
