## [Decision Letter]

**Acceptance summary:**

This paper provides novel and provocative data that will be of interest to the field.

**Decision letter after peer review:**

Thank you for submitting your article "Different dendritic domains of the GnRH neuron underlie the pulse and surge modes of GnRH secretion" for consideration by *eLife*. Your article has been reviewed by three peer reviewers, one of whom is a member of our Board of Reviewing Editors, and the evaluation has been overseen by Catherine Dulac as the Senior Editor. The reviewers have opted to remain anonymous.

The reviewers have discussed the reviews with one another and the Reviewing Editor has drafted this decision to help you prepare a revised submission.

Summary:

Studies in the past decade have demonstrated a curious structure of GnRH neurons where the distal dendrite has partly axon-like properties ("the dendron"). This impressive work has generated some intriguing questions about the control of the gonadotropic axis. Among them is the relative role of different GnRH neuron compartments for the distinct modes of LH secretion, leading to the proposal that the distal dendrite is ideally situated and equipped to integrate upstream input that may constitute the GnRH pulse generator (rather than being intrinsically generated by GnRH neuronal activity). Wang and colleagues describe results from a series of studies investigating the distinct physiological roles of distinct dendritic segments in GnRH neurons.

Previous work in other systems (e.g. Govindaiah and Cox, 2004) has shown how the dendrite can act as a separate, isolated compartment from the rest of the neuron. However, the work on the GnRH system takes this concept a significant step further in showing that this compartmentalized activity can underlie functionally distinct modes of operation (pulse vs. surge). They use a combination of expansion microscopy, chemo- and optogenetic approaches and provide data that activity in proximal dendrites and distal dendrites have distinct responses. They report that (1) GnRH neurons have more synapses on their distal (compared to proximal) dendrites; (2) excitability of distal dendrite controls both pulsatile and surge mode of LH secretion; (3) excitability of soma/proximal dendrite controls surge mode of LH secretion. There are, however, concerns about the design of the study and controls that need to be addressed in order to support the conclusions.

Essential revisions:

Given that CNO diffuses readily in the body (as proved – multiply in the field, and also in this paper – by its effectiveness in the brain when it is given i.p.), what is the evidence or rationale that CNO injected into MBH stays locally in the MBH rather than spreads broadly? Even if CNO binding to DREADD receptor was locally confined (which seems highly improbable), is there any evidence that CNO-induced inhibition stays confined to distal dendron (this also seems unlikely given that DREADD receptor is coupled to diffusible intracellular messengers)? Given that conclusions of the authors are based on the assumption that CNO action is spatially restricted, this seems an important concern.

The chemogenetic approach is powerful, but has been under some criticism recently, with concerns about specificity and sensitivity. Missing from this study is an in vitro validation to demonstrate that application of CNO indeed hyperpolarizes GnRH neurons. Were CNO controls included to exclude effects of metabolism of CNO causing effects? In particular see Gomez et al., 2017.

In Figure 2G, it looks as if the peak was delayed in CNO-treated wild-types. Likewise, even though the CNO-treated tg mouse is largely flat in its curve, it looks as if there is a small increase at 20-21h vs at 19h. Can the authors please comment on this? I note also that in Figure 2H-M no wildtype animals injected with CNO were included as controls.

The optogenetic inhibition experiments have been validated in slices. But the effect seems quite modest. If I read Figure 5—figure supplement 1F correctly, this manipulation decreases firing frequency from ca. 2 to 1 Hz. Indeed, full suppression is difficult to accomplish with continuous illumination owing to the properties of Arch, but in the end, this experiment is difficult to evaluate, since the decrease in firing appears so limited. Can the authors please address how they believe this manipulation, which seems likely to more change the pattern of output of GnRH, rather than dramatically alter the amount of GnRH release, affects LH secretion?

Further issues with controls: blue light was used as a control for the absence of effect of yellow light upon Arch activation in the rPOA on pulsatility (Figure 5F-I and J-M). But this control seems more relevant to include when assessing the effect on surge (to show that blue light does not abolish surge), i.e. to complement Figure 5D-E.

Did the authors perform a negative control (laser in mice without opsins)? E.g. see Owen et al., 2019.

---

## [Author Response]

Essential revisions:Given that CNO diffuses readily in the body (as proved – multiply in the field, and also in this paper – by its effectiveness in the brain when it is given i.p.), what is the evidence or rationale that CNO injected into MBH stays locally in the MBH rather than spreads broadly? Even if CNO binding to DREADD receptor was locally confined (which seems highly improbable), is there any evidence that CNO-induced inhibition stays confined to distal dendron (this also seems unlikely given that DREADD receptor is coupled to diffusible intracellular messengers)? Given that conclusions of the authors are based on the assumption that CNO action is spatially restricted, this seems an important concern.

Indeed, it is critical that the sphere of CNO action is spatially restricted. To our knowledge, the first to undertake a robust assessment of this issue was Stachniak and colleagues (Neuron, 2014) where they showed that intracerebral injections of 3 µM CNO into the paraventricular nucleus had a radius of action in the order of 3-50 µm. While we have not performed similar experiments ourselves, we expect that the 6 µM CNO we injected into the POA would have much the same spread. Indeed, it is clear from our experiments that CNO injected into the POA does not reach sufficient concentrations in the MBH; CNO injected into the POA has no effect on LH pulses whereas CNO injected into the MBH blocks LH pulses. Thus, we feel confident that there is not substantial diffusion of effective CNO from one site to another. We would also note that the amount of CNO injected peripherally (0.15 micromoles) is ~10,000x greater than that injected into the POA or MBH (12 picomoles).

The chemogenetic approach is powerful, but has been under some criticism recently, with concerns about specificity and sensitivity. Missing from this study is an in vitro validation to demonstrate that application of CNO indeed hyperpolarizes GnRH neurons. Were CNO controls included to exclude effects of metabolism of CNO causing effects? In particular see Gomez et al., 2017.

We had just begun to undertake the in vitro validation requested when the COVID19 pandemic struck and all work was suspended. To date we only have a single recording that shows the expected suppression of GnRH neuron firing by CNO (trace shown in Author response image 1) but it is unclear when this work can resume. Many studies have shown suppressive effects of CNO on neural activity both in vivo and vitro (see Smith et al., 2016 for recent review) and we do not believe this is a contentious enough issue to hold up publication of the study. We would be willing to ascribe to the COVID19 *eLife* policy of completing this control when able and posting the result.

The potential effects of clozapine are certainly something to be wary of with this approach. However, we note that CNO given to wild-type, AAV-injected controls has no effect on the LH surge (Figure 2D) indicating that clozapine (or indeed CNO itself) has no impact upon this neural network in our treatment protocol. Other studies in our laboratory (Coutinho et al., 2019) have also shown that intraperitoneal CNO has no effects on pulsatile LH secretion in wild-type mice. Thus, CNO and its metabolites have no effect on either the pulsatile or surge patterns of LH secretion in the absence of the CNO-gated receptor. We have now included reference to this important point in the manuscript.

**Author response image 1. sa2fig1:** Cell-attached recording of a GnRH neuron from an AAV-injected mouse showing the effect of 1 μM CNO on firing.

In Figure 2G, it looks as if the peak was delayed in CNO-treated wild-types. Likewise, even though the CNO-treated tg mouse is largely flat in its curve, it looks as if there is a small increase at 20-21h vs at 19h. Can the authors please comment on this? I note also that in Figure 2H-M no wildtype animals injected with CNO were included as controls.

There are no significant differences between the LH surge profiles of saline- and CNO-treated wild-type mice at any of the time points indicating that there is no effect of CNO alone on the surge mechanism. Regarding the very small non-significant rise at 20 and 21h in CNO-treated tg mice (Figure 2G), this is due to two of the eleven mice (orange and teal traces) showing a very small late increase in LH. In neither case would this be considered an LH surge (see profiles C-E). As shown in Figure 2G there is a highly significant (P<0.001) complete ablation of peak LH surge secretion in CNO-treated tg mice compared to all three controls. Regarding the pulses, as noted above, we have previously demonstrated that intraperitoneal CNO has no effects on pulsatile LH secretion and this is now noted.

The optogenetic inhibition experiments have been validated in slices. But the effect seems quite modest. If I read Figure 5—figure supplement 1F correctly, this manipulation decreases firing frequency from ca. 2 to 1 Hz. Indeed, full suppression is difficult to accomplish with continuous illumination owing to the properties of Arch, but in the end, this experiment is difficult to evaluate, since the decrease in firing appears so limited. Can the authors please address how they believe this manipulation, which seems likely to more change the pattern of output of GnRH, rather than dramatically alter the amount of GnRH release, affects LH secretion?

This is a valid point and we agree with the reviewer that the electrophysiological study shows that the main effect of Arch activation over a prolonged period is to control the pattern of firing rather than suppress activity completely. Indeed, with the 10s on 10s off protocol every 30 min we could reasonably expect GnRH neuron firing to follow this pattern (i.e. an overall 25% decrease) and yet the surge is clearly abolished. This is now raised in the Discussion emphasizing the potential importance of the pattern of GnRH neuron firing.

Further issues with controls: blue light was used as a control for the absence of effect of yellow light upon Arch activation in the rPOA on pulsatility (Figure 5F-I and J-M). But this control seems more relevant to include when assessing the effect on surge (to show that blue light does not abolish surge), i.e. to complement Figure 5D-E.

We have now undertaken the blue light control experiment during the LH surge as requested. The new Figure 5E shows that OVX+E+P GnRH-Cre^+/-^;FLEX-Rosa-CAG-Arch-GFP^+/+^ mice exhibit normal LH surges with bilateral 473nm blue light illumination in rPOA (N=12).

Did the authors perform a negative control (laser in mice without opsins)? E.g. see Owen et al., 2019.

Yes, we originally showed that yellow light illumination had no effect on the LH surge in wild-type mice (Figure 5—figure supplement 2C-E). To address LH pulses, we have now undertaken further control experiments and demonstrate that this laser illumination strategy similarly does not alter pulsatile LH secretion (Figure 5—figure supplement 2F-M).